# LLM-driven Hateful Meme Detection via Cross-modal Memorizing and Self-rejection Training

## Abstract

Hateful meme detection (HMD) is critical for determining whether online multimodal content carries harmful information, which plays a pivotal role in maintaining a harmonious internet ecosystem. HMD is predominantly viewed as a multimodal task, where the harmful message in memes is expressed through the information conveyed by the combination of visual and text content (e.g., the contradictions between them) rather than that from one modality. Thus, effective modeling and smooth integration of multimodal information are crucial for achieving promising HMD performance. Current research on HMD conventionally models visual and text data independently, subsequently aligns and merges these multimodal features for HMD predictions. However, existing studies face challenges in identifying hateful information that derives from the complementarities or contradictions between image and text, where in most cases neither image nor text alone carries explicit hateful information. Moreover, these studies do not leverage the capabilities of large language models (LLMs), which have been demonstrated effective in cross-modal information processing. Therefore in this paper, we propose a multimodal approach for HMD following the encoding-decoding paradigm with using LLM and a memory module enhanced by self-rejection training. Particularly, the memory module learns appropriate relationships between image and text that lead to hateful memes, where the resulted information is fed into the LLM and accompanied with visual and text features to predict HMD labels. Self-rejection training performs a discriminative learning according to memory outputs and enhances the memory module to improve HMD. We evaluate our approach on English and Chinese benchmark datasets, where it outperforms strong baselines, demonstrating the effectiveness of all components in it and our model design.[1]

*Note*: This paper contains examples of hate speech.

## 1 Introduction

Multimodal memes are typically characterized as images infused with text that propagate from one individual to another, which have become a widespread form of expression on social media platforms (Kiela et al., 2020; Gomez et al., 2020) and a certain amount of them convey hateful information so that are potential in causing negative emotions and further harm to Internet users. Consider that memes on Internet are fast and widely spread, detecting hateful memes with artificial intelligence (AI) is of great importance for cyberspace maintenance. Therefore, advanced cross-model understanding techniques are expected in doing so to fulfill the requirement of in-time and precise hateful meme detection (HMD), where multimodal modeling becomes particularly pronounced in this task. Figure 1 shows three comparing examples that emphasize the significance of synchronous visual and text understanding, where Figure 1 (a) displays hateful memes and the Figure 1 (b) and (c) present non-hateful ones, illustrating that different image and text combinations delivering opposite attitude tendencies. Hence, relying solely on modeling images or text proves insufficient for HMD, where a more robust approach necessitates enhanced unified modeling of both modalities.

Existing approaches utilize advanced visual and text encoders (such as CLIP (Radford et al., 2021), Flamingo (Alayrac et al., 2022), FLAVA (Singh et al., 2022), and SLIP (Mu et al., 2022), etc.) to

---

[1]The code and model will be released in the final version of this paper.

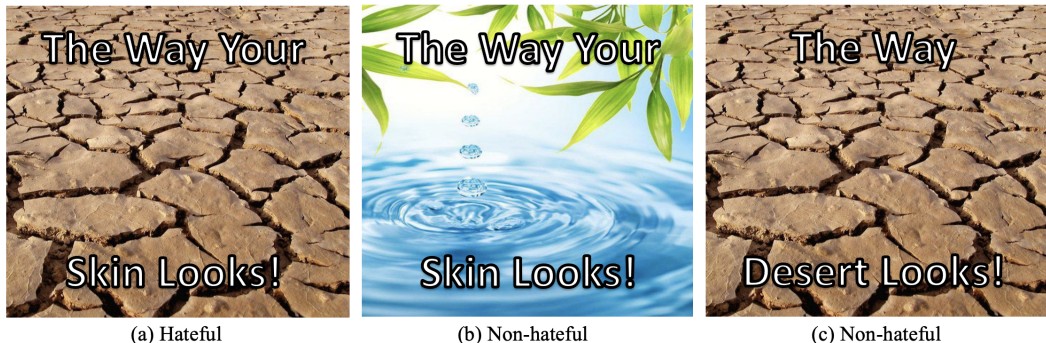

| (a) Hateful | (b) Non-hateful | (c) Non-hateful |

Figure 1: Examples compare hateful meme (a) with non-hateful ones (b) and (c). Herein, (b) and (c) share the same text and visual content with (a), respectively, but do not show hateful information as (a) does, which shows that the combination of different images and texts leads to opposite attitudes.

extract multimodal features, and subsequently align or fuse them by vector concatenation, outer production, or attentions to improve HMD (Kiela et al., 2019; Li et al., 2019; Radford et al., 2021; Goyal et al., 2022; Nandakumar, 2022; Koutlis et al., 2023). These models successfully identify hateful memes where images or texts present explicit biases but are unable to effectively recognize hateful information derived from complementarities or contradictions between visual and textual content other than images or text alone. Although there are efforts in utilizing additional resources or using model ensemble to improve HMD Muennighoff (2020); Lippe et al. (2020); Sandulescu (2020); Velioglu & Rose (2020); Zhu (2020); Cao et al. (2023), they mainly enhance the generalization ability through more training data or take advantage of different models, without touching the essential mechanism that leads to hateful information. In addition, existing approaches omit the chance to leverage large language models (LLMs), such as MiniGPT-4 (Zhu et al., 2023) and LLaVA (Liu et al., 2023a), which have proven effective in a broad range of cross-modal tasks. Therefore, it is expected to further advance HMD approaches with rational and efficient solutions to model appropriate relationships between visual and text semantics.

In this paper, we propose a multimodal approach with LLM to enhance HMD through self-rejection training. Our approach learns the relationship between visual and textual content that leads to hateful memes through a memory module, which is pipelined with another two components, a visual encoder capturing image features and an LLM predicting HMD labels. We further propose a self-rejection training procedure to optimize the memory module by rectifying correlation vectors from the memory against direct image-text matching results, so as to better capture essential task-specific information to improve HMD. Evaluations on benchmark datasets demonstrate that our approach outperforms strong baselines and existing approaches, emphasizing the superiority of our memory and self-rejection training for HMD.

## 2  THE APPROACH

Figure 2 illustrates the framework of our approach, where the memory-based HMD pipeline and the self-rejection training process are presented at the top and bottom of the figure, respectively. Overall, the pipeline follows the convention of existing studies to regard HMD as a multimodal classification task, which predicts a label $\mathcal{Y}$ based on the image $\mathcal{I}$ and embedded text $\mathcal{X}$ in a given meme $(\mathcal{I}, \mathcal{X})$. Moreover, the proposed self-rejection training enhances the our approach by effectively aligning memories with crucial information (e.g., contradictions between visual and text content) that leads to hateful memes. The following text illustrates the pipeline and self-rejection training in details.

### 2.1  THE HMD PIPELINE

The pipeline of our approach consists of three essential components: visual encoding, cross-modal memorizing, and LLM prompting. Specifically, the visual encoding process ($f_{ve}$) extracts salient features from the input image; the cross-modal memory module ($f_m$) encodes the correlation between visual and text features; the LLM prompting ($f_d$) utilizes the multimodal information to predict the final label $\widehat{\mathcal{Y}}$. Therefore, our approach is formulated by

$$\widehat{\mathcal{Y}} = f_d(f_{ve}(\mathcal{I}), f_m(f_{ve}(\mathcal{I}), \mathcal{X}), \mathcal{X}, p) \tag{1}$$

where $p$ denotes the prompt for LLM. In the following text, we present each component in detail following the aforementioned processing sequence.

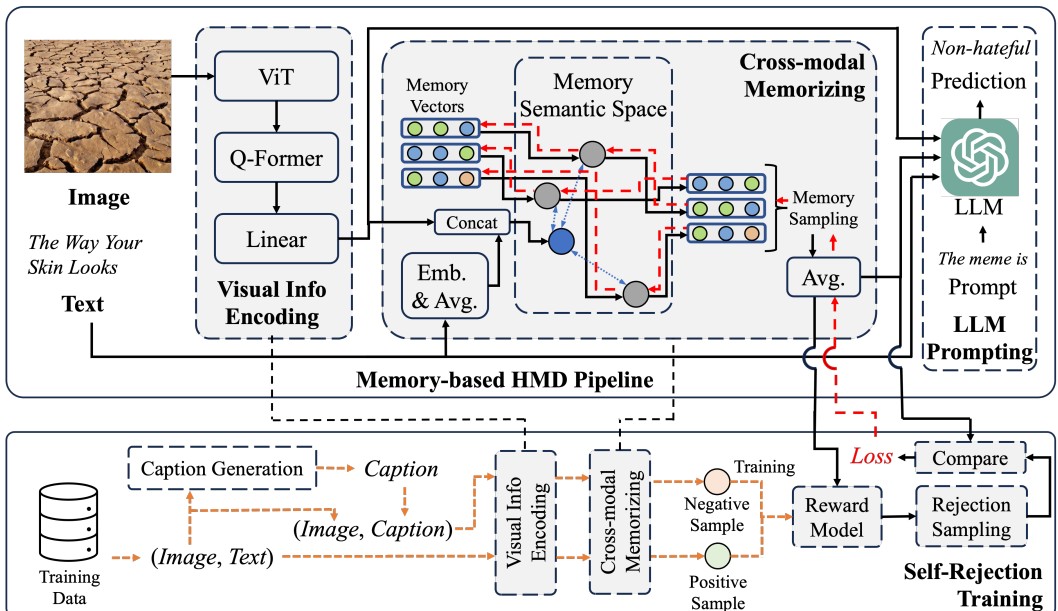

Figure 2: The overall architecture of our approach. The top and bottom parts illustrate our memory-based HMD pipeline and the self-rejection training processes, respectively. The process of training the reward model is illustrated in orange arrows in the self-rejection training part. The dashed red arrows illustrate the process of using loss to optimize the memory module through self-rejection training. Moreover, visual encoding and cross-modal memorizing are shared by the entire HMD pipeline and self-rejection training, which is indicated by the dashed black lines.

**Visual Encoding**   Our approach to encoding visual signals follows the procedure of BLIP2 (Li et al., 2023) with three components: the vision Transformer $f_v$ (Dosovitskiy et al., 2021), the Q-Former $f_q$ (Li et al., 2023), and a linear projection layer. The three modules are sequentially interconnected to extract visual feature $\mathbf{v}$ from the input meme $\mathcal{I}$ through

$$\mathbf{v} = f_{ve}(\mathcal{I}) = Linear(f_q(f_v(\mathcal{I}))) \tag{2}$$

In our approach, the vision Transformer $f_v$ distills crucial visual features from the meme, and the Q-Former $f_q$ translates these features into a textual semantic space, then finally, the linear projection layer transforms the resulted representation into latent vectors $\mathbf{v}$, ensuring alignment with the dimensional space of hidden states in the subsequent module.

**Cross-modal Memorizing**   The memory module is designed to capture crucial information, i.e., correlation between visual and text features leading to hateful memes. In doing so, we propose a memory matrix represented by $N$ vectors (denoted by $[\mathbf{m}_1, \cdots, \mathbf{m}_N]$), where each memory vector can be interpreted as a potential aspect resulting in hateful information. Memory searching and memory sampling are the main steps in this module, with their details illustrated as follows.

Memory searching locates relevant memory vectors according to the encoded multimodal information and assigns appropriate weights to them. For the input multimodal information, in addition to the visual encoding, we obtain the textual representation by averaging the embeddings of all tokens in the input text by $\mathbf{t} = \frac{1}{U} \sum_{u=1}^{U} \mathbf{e}_u$, with each $\mathbf{e}_u \in [\mathbf{e}_1 \cdots \mathbf{e}_U]$ ($U$ refers to total token number) denoting embedding for its corresponding token. Then, we concatenate visual and text features and obtain the multimodal feature $\mathbf{x}_{vt} = \mathbf{v} \oplus \mathbf{t}$. Afterwards, we compute the weight $w_n$ that measures the semantic similarity between the $n$-th memory vector $\mathbf{m}_n$ and $\mathbf{x}_{vt}$ by

$$w_n = \frac{\exp(\mathbf{x}_{vt} \cdot \mathbf{W}_m \cdot \mathbf{m}_n)}{\sum_{n=1}^{N} \exp(\mathbf{x}_{vt} \cdot \mathbf{W}_m \cdot \mathbf{m}_n)} \tag{3}$$

where $\mathbf{W}_m$ is a trainable parameter matrix to align $\mathbf{m}_n$ and $\mathbf{x}_{vt}$. Finally, we rank all memory vectors in descending order based on their weights and select the top $N'$ vectors (denoted as $\mathbf{m}_{n_1} \cdots \mathbf{m}_{n_{N'}}$) as the relevant vectors for later processing.

Memory sampling further processes memory vectors and outputs a correlation vector $\mathbf{x}_m$ that carries the essential correlation information between visual and text features for later steps. In detail, we normalize the weights of the relevant vectors and randomly select one from $\mathbf{m}_{n_1} \cdots \mathbf{m}_{n_{N'}}$ based on their weights, where higher weights lead to better chance to be selected. Subsequently, we perform

the sampling process $M$ times[2] and obtain a vector list $\mathbf{m}_{n_1} \cdots \mathbf{m}_{n_M}$, with repetition of the same vector allowed. We then average the list and obtain the output correlation vector $\mathbf{x}_m$ by

$$\mathbf{x}_m = \frac{1}{M} \sum_{m=1}^{M} \mathbf{m}_{n_m} \tag{4}$$

where $\mathbf{x}_m$ is used in self-rejection training for further enhancing the memory module as well as the output of the memory module for later HMD prediction process.

**LLM Prompting**   Existing studies on LLM have demonstrated the significant impact of prompting on model performance (Brown et al., 2020; Lester et al., 2021; Ouyang et al., 2022; Liu et al., 2022). For better prompting, we use the visual feature $\mathbf{v}$ and the correlation vector $\mathbf{x}_m$ as soft prompts to guide our LLM for HMD. Specifically, we feed $\mathbf{v}$, $\mathbf{x}_m$, as well as the original text $\mathcal{X}$, into the LLM to determine the label $\widehat{\mathcal{Y}}$, i.e., *hateful* or *non-hateful*. In doing so, a prompt $p$ is required to instruct the LLM to process the input and predict the HMD label.[3] Therefore, we feed $\mathbf{v}, \mathbf{x}_m, \mathcal{X}, p$ into our LLM (i.e., Vicuna Chiang et al. (2023)) and obtain the hidden vector $\mathbf{h}$ from its last layer by

$$\mathbf{h} = LLM(\mathbf{v}, \mathbf{x}_m, \mathcal{X}, p) \tag{5}$$

Afterwards, we compute the HMD score from the vector $\mathbf{h}$ by

$$s_h = \mathbf{e}_h \cdot \mathbf{h}, \quad s_{nh} = \mathbf{e}_{nh} \cdot \mathbf{h} \tag{6}$$

where $\mathbf{e}_h$ and $\mathbf{e}_{nh}$ denote trainable embeddings corresponding to the *hateful* and *non-hateful* labels and leading to their scores, $s_h$ and $s_{nh}$, respectively. Finally, we compare $s_h$ and $s_{nh}$ and output the final prediction $\widehat{\mathcal{Y}}$ according to which one is higher.

## 2.2   SELF-REJECTION TRAINING

In this process, we further assess correlation vectors $\mathbf{x}_m$ to evaluate whether they contain crucial information (e.g., contradictions between image and text) that lead to hateful information and thus adjust the memory module accordingly so that ensuring it iteratively produces better output. The self-rejection training process consists of two steps: reward model training and rejection sampling, with their details elaborated in the following text.

**Reward Model Training**   The reward model measures the effectiveness of the encoded correlation vector for representing visual and text features in detecting hateful meme. Therefore, we train the reward model by distinguishing such correlation information embedded in the vectors that are relevant or irrelevant to HMD, so as to ensure the model to assign high scores to those ones helpful to the task. Therefore, we treat HMD-related and irrelevant cross-modal instances as positive and negative samples, respectively, to train the reward model. In doing so, we randomly select an instance, i.e., image-text pair $(\mathcal{I}_r, \mathcal{X}_r)$, from the training data and treat it as a positive sample. Then we generate a caption $\mathcal{C}_r$ for the image in this instance and combine it with the image to form a negative sample $(\mathcal{I}_r, \mathcal{C}_r)$. Later we apply the same visual encoding and the memory module in our HMD pipeline to compute the correlation vectors for the positive and negative samples by

$$\mathbf{v}_m^{pos} = f_m(f_{ve}(\mathcal{I}_r), \mathcal{X}), \quad \mathbf{v}_m^{neg} = f_m(f_{ve}(\mathcal{I}_r), \mathcal{C}) \tag{7}$$

to obtain positive and negative correlation vectors $\mathbf{v}_m^{pos}$ and $\mathbf{v}_m^{neg}$, respectively. Finally, we feed $\mathbf{v}_m^a$ and $\mathbf{v}_m^r$ to the reward model $f_r$, which is a multi-layer perceptron, and compute the scores (denoted as $s_a$ and $s_r$, respectively) for the vectors by

$$s_{pos} = sigmoid(f_r(\mathbf{v}_m^{pos})), \quad s_{neg} = sigmoid(f_r(\mathbf{v}_m^{neg})) \tag{8}$$

and compute the loss $\mathcal{L}_r$ to optimize the reward model by

$$\mathcal{L}_r = -log s_{pos} - log(1 - s_{neg}) \tag{9}$$

---

[2]We perform the random selection to facilitate the self-rejection training process illustrated in Section 2.2

[3]An example prompt is "*Is the meme hateful or non-hateful?*"

| | **HMC** | | **Memeplate** | | | |
| | Dev | | Dev | | Test | |
| | ACC | AUROC | ACC | F1 | ACC | F1 |
|---|---|---|---|---|---|---|
| **Base (BLIP2)** | $71.36_{\pm 0.24}$ | $81.05_{\pm 0.20}$ | $51.45_{\pm 0.21}$ | $45.14_{\pm 0.26}$ | $51.72_{\pm 0.24}$ | $45.51_{\pm 0.22}$ |
| **+M** | $72.06_{\pm 0.22}$ | $81.86_{\pm 0.25}$ | $52.81_{\pm 0.21}$ | $46.08_{\pm 0.24}$ | $52.87_{\pm 0.19}$ | $46.23_{\pm 0.23}$ |
| **+SRT** | $72.44_{\pm 0.19}$ | $82.19_{\pm 0.20}$ | $53.01_{\pm 0.24}$ | $46.44_{\pm 0.23}$ | $53.29_{\pm 0.22}$ | $46.84_{\pm 0.20}$ |
| **+M+SRT** | $^*\mathbf{72.70}_{\pm 0.20}$ | $^*\mathbf{82.88}_{\pm 0.21}$ | $^*\mathbf{53.47}_{\pm 0.18}$ | $^*\mathbf{46.92}_{\pm 0.22}$ | $^*\mathbf{53.63}_{\pm 0.21}$ | $^*\mathbf{47.13}_{\pm 0.20}$ |
| **Base (LLM)** | $76.24_{\pm 0.30}$ | $84.46_{\pm 0.22}$ | $53.65_{\pm 0.24}$ | $47.84_{\pm 0.28}$ | $55.42_{\pm 0.21}$ | $49.03_{\pm 0.19}$ |
| **+M** | $77.08_{\pm 0.24}$ | $85.44_{\pm 0.20}$ | $55.10_{\pm 0.18}$ | $48.98_{\pm 0.22}$ | $56.07_{\pm 0.20}$ | $49.43_{\pm 0.26}$ |
| **+SRT** | $77.46_{\pm 0.18}$ | $85.69_{\pm 0.21}$ | $55.31_{\pm 0.20}$ | $49.34_{\pm 0.24}$ | $56.39_{\pm 0.19}$ | $49.77_{\pm 0.22}$ |
| **+M+SRT** | $^*\mathbf{78.08}_{\pm 0.24}$ | $^*\mathbf{86.84}_{\pm 0.19}$ | $^*\mathbf{56.52}_{\pm 0.17}$ | $^*\mathbf{50.07}_{\pm 0.23}$ | $^*\mathbf{56.83}_{\pm 0.20}$ | $^*\mathbf{50.34}_{\pm 0.19}$ |

Table 1: The performance (i.e., the average and standard deviation of different evaluation metrics) of various models on the development and test sets of HMC and Memeplate datasets. "Base (BLIP2)" stands for BLIP2 models used only for HMC and Memeplate; "Base (LLM)" stands for MiniGPT-4 and Ziya–BLIP2-Visual models used only for HMC and Memeplate, respectively. "M" and "SRT" are abbreviations of the memory module and self-rejection training, respectively. $^*$ marks the results where improvements are statistically significant at $p \leq 0.05$ level over all baselines.

**Rejection Sampling**  This process includes two steps, namely, correlation vector scoring and rejection sampling fine-tuning, which are elaborated as follows. In correlation vector scoring, for a particular input meme $(\mathcal{I}, \mathcal{X})$, we run sampling in our memory module $T$ times and get $T$ correlation vectors, denoted as $\mathbf{x}_m^1 \cdots \mathbf{x}_m^T$. Then we feed all correlation vectors to the reward model $f_r$ and compute the score for each of them. In rejection sampling fine-tuning, we select the correlation vector with the highest score (denoted as $\mathbf{x}_m^*$) and use it as the gold standard to assess whether the correlation vector from the memory module is good enough to carry essential task-specific information for HMD. Finally we compute the loss

$$\mathcal{L}_{rsft} = |\mathbf{x}_m^* - \mathbf{x}_m| \tag{10}$$

to update the memory module with $|\cdot|$ denoting the norm of a vector and $\mathbf{x}_m$ obtained from Eq. (4).

## 3 EXPERIMENT SETTINGS

### 3.1 DATASETS

We employ two datasets in our experiments, namely, HMC dataset (Kiela et al., 2020) and Memeplate (Li et al., 2022). The HMC is an English dataset including 10,000 instances of memes and their corresponding text. Memeplate is a Chinese dataset for multimodal humor recognition, which contains 203 templates and 5,184 memes with manually annotated humor levels. We use this dataset to further comprehensively evaluate the capability of our approach for HMD, since humor recognition is also a challenging classification task that necessitates a deep understanding of both visual and text elements. For both datasets, we use their official training, development, and test data split.[4] Note that, since the label of the test set of HMC is not publicly available, we follow existing studies (Radford et al., 2021; Goyal et al., 2022; Singh et al., 2022; Cao et al., 2023; Koutlis et al., 2023), to evaluate all models on its development set.

### 3.2 BASELINES

In our experiment, we employ MiniGPT-4 as the backbone model (which is based on BLIP2 (Li et al., 2023)) for the English task, which is recognized as a prominent multimodal LLM with promising performance across numerous multimodal tasks. For Chinese, we use Ziya-BLIP2-Visual (Zhang et al., 2022) that employs the same architecture as MiniGPT-4. We also try settings with small language models, i.e., GPT-2 (Radford et al., 2019) following the same BLIP2 architecture. To compare with the proposed approach, we run experiments with the following three baseline models: (1) **BLIP2, MiniGPT-4 and Ziya (Base)**, which is the original version of BLIP2, MiniGPT-4 or Ziya-BLIP2-Visual; (2) the **Base+M** model where the base models in (1) are enhanced by the proposed memory module, which includes visual information encoding, cross-modal memorizing, and LLM prompting, and the memory is optimized by the cross-entropy loss from comparing the model prediction and the gold standard for the task. (3) the **Base+SRT** model where we directly use self-rejection sampling to enhance the base models by concatenating visual and text features (i.e., $\mathbf{x}_{vt}$) to form the correlation vector (i.e., $\mathbf{x}_m$), and randomly set 33% values in $\mathbf{x}_m$ to zero to facilitate self-rejection training.

---

[4]We report the statistics of the dataset in Appendix A.

| | ACC | AUROC |
|---|---|---|
| Muennighoff (2020) | - | 81.56 |
| Velioglu & Rose (2020) | 70.93 | 75.21 |
| Lippe et al. (2020) | - | 77.39 |
| Radford et al. (2021) | - | 77.30 |
| Goyal et al. (2022) | - | 73.40 |
| Nandakumar (2022) | - | 81.55 |
| Singh et al. (2022) | - | 76.70 |
| Cao et al. (2023) | 72.98 | 82.45 |
| Koutlis et al. (2023) | 73.60 | 80.10 |
| △Liu et al. (2023a) | 76.20 | 84.57 |
| **Ours** | **78.08** | **86.84** |

Table 2: Comparison of our approach with existing studies on HMC. "△" marks our own runs of multi-modal systems with LLMs. We report the average performance of our approach only on the development set since the gold standard labels of the test set are not publicly available.

| | ACC | F1 |
|---|---|---|
| *RB + ResNet-50 | 51.08 | 45.82 |
| *RB + XCiT | 49.32 | 46.18 |
| *RB + BEiT | 52.30 | 45.33 |
| *RB + Faster-RCNN | 50.54 | 43.31 |
| †Yang et al. (2022) | 52.57 | 46.21 |
| †△ Yang et al. (2023) | 55.43 | 48.80 |
| †△ Hu et al. (2023) | 55.08 | 48.97 |
| †△ University (2023) | 55.76 | 49.49 |
| **Ours** | **56.83** | **50.34** |

Table 3: Performance comparison of different models on the test set of Memeplate dataset. Scores marked by "*" and "†" are from Li et al. (2022) and our own runs on this dataset, respectively. "RB" stands for the RoBERTa model; "△" indicates that the multimodal models using LLMs to predict labels.

## 3.3 Implementation Details

The HMD pipeline of our approach for HMC and Memeplate is based upon BLIP2, MiniGPT-4, and Ziya-BLIP2-Visual, utilizing 12, 32 and 40 layers of multi-head attentions with 1.5B, 7B and 13B parameters, respectively. Specifically, the visual encoding and LLM prompting processes in our approach follow the same procedures as that applied in these foundation models. The visual transformer and Q-Former in visual encoding consist of 40 and 12 transformer layers, respectively. In fine-tuning our approach, we alternate between the following two procedures for every 100 steps: (1) updating the parameters of different components in visual encoding, memory module, and LLM using the cross-entropy loss from comparing the predicted labels with gold standards and (2) updating the reward model and the memory module through self-rejection training.[5] For evaluation, we follow existing studies (Kiela et al., 2020; Li et al., 2022; Cao et al., 2023; Koutlis et al., 2023) to use accuracy and AUROC for HMC while accuracy and F1 for Memeplate. We try a series of hyperparameter settings and select the one that yield the best performance on the development set[6] in our final experiments. I.e., the numbers of memory vectors (i.e., $N$) for HMC and Memeplate are 200 and 150, respectively; the relevance memory size (i..e., $M$) and sampling time $T$ are set to 20 and 4, respectively; the learning rate is $1 \times 10^{-6}$ and the batch size is 32 for both datasets. We run baselines and our approach five times with different random seeds and record the average and standard deviation of model performance.

## 4 Results and Analysis

### 4.1 Overall Performance

We run baselines and our approach on HMC and Memeplate datasets and report the average model performance with standard deviations in Table 1. There are following observations. First, our approach consistently outperforms baselines, which indicates the effectiveness of the proposed approach for HMD given that baseline models have already achieved promising performance. Second, adding memory module (i.e., "+M") or self-rejection training (i.e., "+SRT") leads to noticeable improvements over the "Base" model, which illustrates the effectiveness of individual modules to capture correlations between visual and text information to improve model performance. Third, compared with "+M", "+SRT" presents higher performance over different settings, indicating the superiority of discriminative learning on task-specific information. Fourth, our full model with both the memory module and self-rejection training outperforms all baselines, demonstrating the necessity of combining them to further enhance HMD.

We further compare our approach with existing studies and report the results for HMC and Memeplate in Table 2 and Table 3, respectively. It is observed that our approach outperforms previous studies on both datasets, especially the ones using powerful pre-trained multimodal models (Nandakumar, 2022; Singh et al., 2022; Koutlis et al., 2023). The reason behind the observation is that,

---

[5]For self-rejection training with the Base+SRT baseline, we update the parameters of visual encoding and the token embeddings of the input text, so as to deal with the situation of memory module absence.

[6]For HMC, we randomly select 10% of the training data and use it to tune hyper-parameters.

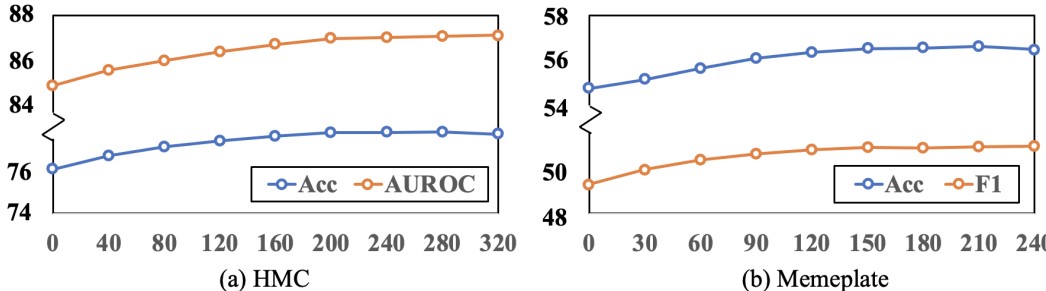

Figure 3: Curves of model performance on the development set of HMC and the test set of Memeplate with respect to different numbers of memory vectors used in the memory module.

| | HMC | | Memeplate | | | |
|---|---|---|---|---|---|---|
| | Dev | | Dev | | Test | |
| | ACC | AUROC | ACC | F1 | ACC | F1 |
| **OP** | $76.78_{\pm 0.26}$ | $84.80_{\pm 0.20}$ | $53.94_{\pm 0.21}$ | $48.39_{\pm 0.24}$ | $55.90_{\pm 0.27}$ | $49.55_{\pm 0.22}$ |
| **Co-Att** | $76.96_{\pm 0.22}$ | $84.91_{\pm 0.23}$ | $54.57_{\pm 0.18}$ | $48.50_{\pm 0.22}$ | $55.90_{\pm 0.24}$ | $49.19_{\pm 0.21}$ |

Table 4: Performance of different models with the memory module in our approach replaced by outer product operation (OP) and Co-attention mechanism (Co-Att).

| | HMC | | Memeplate | | | |
|---|---|---|---|---|---|---|
| | Dev | | Dev | | Test | |
| | ACC | AUROC | ACC | F1 | ACC | F1 |
| **M** | $77.70_{\pm 0.21}$ | $88.57_{\pm 0.20}$ | $56.31_{\pm 0.14}$ | $49.64_{\pm 0.20}$ | $56.60_{\pm 0.22}$ | $49.83_{\pm 0.19}$ |
| **LoRA** | $77.96_{\pm 0.24}$ | $88.75_{\pm 0.23}$ | $56.40_{\pm 0.19}$ | $49.81_{\pm 0.26}$ | $56.77_{\pm 0.20}$ | $50.07_{\pm 0.23}$ |

Table 5: Performance of our approach with different fine-tuning strategies. "M" stands for the setting where we only fine-tune the memory module in the pipeline and fix the parameters in the visual encoding and the LLM; "LoRA" refers to that we use LoRA to fine-tune the LLM, where the parameters in the visual encoding and the memory module are also updated simultaneously.

these multimodal models generally perform HMD in the same way as image captioning, which focuses on the content shared by image and text rather than their correlations that lead to other (i.e., hateful) information. On the contrary, our approach correctly distinguish such correlation with our particular model design so that leads to better performance.

## 4.2 EFFECT OF THE MEMORY MODULE

The memory matrix in the proposed memory module represent the semantic space for the correlation between visual and text features for the specific task. Therefore, it is necessary to investigate the effect of the matrix, especially its vector numbers (i.e., $N$), on HMD performance. In doing so, we run experiments with different $N$ on HMC and Memeplate, where the curves of model performance with respect to $N$ are illustrated in Figure 3, with following observations. First, when $N$ is relatively small, increasing the number of memory vectors leads to noticeable improvements of model performance, which is not surprising since a smaller $N$ corresponds to a restricted space in capturing essential correlation information. Second, with $N$ grows, the performance converges, demonstrating that once the memory vectors cover enough information of cross-modal correlation that results in hateful information, adding more vectors has limited effect to further benefit HMD.

In addition, to better illustrate the effect of memory module when it coordinates with self-rejection training, we run two additional approaches where the memory module in our approach is replaced by outer product operation (OP) and co-attention (Co-Att) (Lu et al., 2016).[7] We report the experimental results of the two models in Table 4, and observe that the two models achieve worse performance compared with the "+SRT" baseline as well as our full model as that shown in Table 1, which demonstrates the effectiveness of our design with memory and self-rejection training. This observation further confirms the superiority of modeling task-specific correlation for HMD, since OP and Co-Att are widely used to align and fuse multimodal features in tasks such as image captioning and proved to be effective in modeling the semantics shared by multimodalities, which is different from the correlation information between visual and text features in our task.

---

[7]For the two additional models, in detail, for OP, we compute the outer product of the visual and text features, flatten the resulting matrix, and use the resulted vector as the correlation vector $\mathbf{x}_m$; for Co-Att, we utilize co-attention to fuse multimodal features and directly regard the output as the correlation vector.

| ID | Dataset | Prompt |
|---|---|---|
| 1 | HMC | *The meme is* |
| | Memeplate | *The humor level of the meme is* |
| 2 | HMC | *You are asked to predict whether the meme is hateful or non-hateful based on the given visual and text features. The meme is* |
| | Memeplate | *You are asked to predict the humor level (ranging from 1-3) of the meme based on the given visual and text features. The humor level of the meme is* |
| 3 | HMC | *Is the meme hateful or non-hateful?* |
| | Memeplate | *What is the humor level of the meme?* |
| 4 | HMC | *You are asked to predict whether the meme is hateful or non-hateful based on the given visual and text features. Is the meme hateful or non-hateful?* |
| | Memeplate | *You are asked to predict the humor level (ranging from 1-3) of the meme based on the given visual and text features. What is the humor level of the meme?* |

Table 6: Prompts used to investigate the robustness of our approach.

| Prompt ID | HMC | | Memeplate | | | |
|---|---|---|---|---|---|---|
| | Dev | | Dev | | Test | |
| | ACC | AUROC | ACC | F1 | ACC | F1 |
| 1 | $78.00_{\pm0.23}$ | $86.74_{\pm0.20}$ | $56.48_{\pm0.19}$ | $50.00_{\pm0.21}$ | $56.78_{\pm0.20}$ | $50.27_{\pm0.23}$ |
| 2 | $77.96_{\pm0.20}$ | $84.64_{\pm0.21}$ | $56.60_{\pm0.22}$ | $50.13_{\pm0.18}$ | $56.80_{\pm0.21}$ | $50.21_{\pm0.20}$ |
| 3 | $78.08_{\pm0.24}$ | $86.84_{\pm0.19}$ | $56.52_{\pm0.17}$ | $50.07_{\pm0.23}$ | $56.83_{\pm0.20}$ | $50.34_{\pm0.19}$ |
| 4 | $77.92_{\pm0.19}$ | $86.68_{\pm0.21}$ | $55.42_{\pm0.23}$ | $50.10_{\pm0.20}$ | $56.90_{\pm0.23}$ | $50.38_{\pm0.21}$ |

Table 7: Performance of our approach on HMC and Memeplate with the prompts from Table 6.

## 4.3 EFFECT OF FINE-TUNING STRATEGY

Fine-tuning strategies have great impact in model training. To investigate the influence of different strategies, we experiment with two settings: (1) we only update the parameters in the memory module and fix those in the visual encoding and the LLM; (2) we use LoRA (Hu et al., 2021) to fine-tune the LLM and fine-tune all parameters in the visual encoding and the memory module. The results are reported in Table 5, where observations drawn as follows. First, when only the memory parameters are fine-tuned, there is a slight drop in performance compared with full parameter fine-tuning (see Table 1), owing to the reason of potential information mismatch among updated and fixed modules. However, the fact of slightly dropping is a further confirmation of our model design by indicating the power of memory compared with the non-memory baseline. Second, the performance of LoRA fine-tuning was comparable to the full-parameter fine-tuning, which demonstrates the robustness and flexibility of our approach working with various effective fine-tuning techniques.

## 4.4 EFFECT OF DIFFERENT PROMPTS

Existing studies demonstrated that different designs on prompting have significant influences on LLM performance Schick & Schütze (2021); Liu et al. (2023b); White et al. (2023). Therefore, we analyze model performance with various prompts and provide insights on the robustness and generalization capabilities of our approach. In doing so, we try different prompts illustrated in Table 6, where the prompts differ from each other from the following two perspectives: (1) with and without task description in the prompt (i.e., prompt 2 vs. prompt 1 and prompt 4 vs. prompt 3), and (2) follow or do not follow question formats (i.e., prompt 3 vs. prompt 1 and prompt 4 vs. prompt 2). We report the performance of our approach with different prompts in Table 7, where our approach works well with various prompts and stabilizes on HMD results, demonstrating the robustness of applying LLM in our approach.

## 4.5 CASE STUDY

In addition to quantitative study, we investigate three similar memes for qualitative analysis. The memes and the prediction from different models, as well as the gold standard, are illustrated in Figure 4, where correct and incorrect predictions are highlighted in green and red colors, respectively.[8] Herein, meme (a) and (b) share the same texts while meme (a) and (c) share the same visual content, leading to that (a) being the hateful one but (b) and (c) are not. By investigating the results, we observe that the three baselines struggle to consistently predict HMD correctly, whereas our full model

---

[8] We present another case in Appendix B.

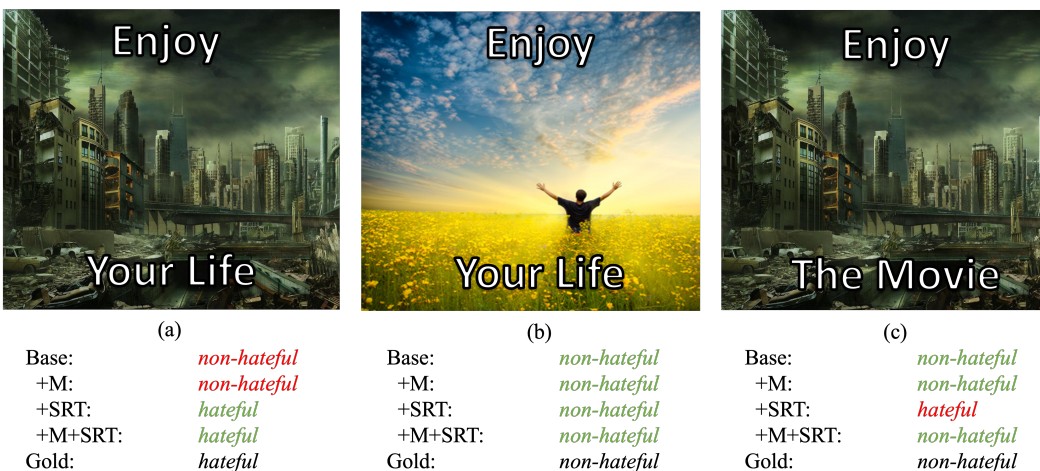

|  | (a) |  | (b) |  | (c) |
|---|---|---|---|---|---|
| Base: | *non-hateful* | Base: | *non-hateful* | Base: | *non-hateful* |
| +M: | *non-hateful* | +M: | *non-hateful* | +M: | *non-hateful* |
| +SRT: | *hateful* | +SRT: | *non-hateful* | +SRT: | *hateful* |
| +M+SRT: | *hateful* | +M+SRT: | *non-hateful* | +M+SRT: | *non-hateful* |
| Gold: | *hateful* | Gold: | *non-hateful* | Gold: | *non-hateful* |

Figure 4: Three memes and predictions of different models on them, with the gold standards also presented. Correct and incorrect labels are highlighted in green and red colors, respectively.

is able to accurately identify all meme types. The reason is similar to that for analyzing OP and Co-Att replacement in §4.2, where the hateful information in memes generally derives from the correlation (i.e., the contradiction relationship in this case) between visual and text features rather than how well the image and text matches. The baselines have their limitations that prevent them from learning such correlation, either lacking particular mechanism to do so or being equipped without effective guidance. In contrast, the memory module and self-rejection training applied in our approach provide a comprehensive solution to learn, weight, and enhance such information so as to better identify hateful information in memes.

## 5 RELATED WORK

HMD is a crucial task for safeguarding the digital sphere from harmful content, which is relevant to tasks such as meme emotion analysis, offensive meme detection, and harmful meme detection (Suryawanshi et al., 2020; Sharma et al., 2020; Pramanick et al., 2021a;b; Kocoń et al., 2021; Sharma et al., 2022a;b; Hakimov et al., 2022). Although hateful memes are often conveyed by both images and texts, some early studies for HMD leverage unimodal approaches, where only one type of modality is used to detect them (Ren et al., 2015; He et al., 2016; Devlin et al., 2019; Kiela et al., 2021). Another stream of research introduces multimodal approaches that combine both image and text encoders for better results, where superior visual and text encoders (such as MMBT (Kiela et al., 2019), ViLBERT (Lu et al., 2019), VisualBERT (Li et al., 2019), CLIP (Radford et al., 2021), Flamingo (Alayrac et al., 2022), FLAVA (Singh et al., 2022), SLIP (Mu et al., 2022)) are used to extract features from images and text, respectively, then they further align or fuse multimodality features with a particular module or operation, such as vector concatenation and attentions (Goyal et al., 2022; Nandakumar, 2022; Koutlis et al., 2023; Hee et al., 2023). To further enhance HMD, model ensemble (Muennighoff, 2020; Lippe et al., 2020; Sandulescu, 2020), additional resources (e.g., extra training data and features) (Velioglu & Rose, 2020; Zhu, 2020), contrastive learning (Liang et al., 2022; Qu et al., 2023), and language model prompting (Cao et al., 2023) are employed to improve the ability to capture multimodality features, where limited attention is paid to model essential relationships between visual and text content that lead to hateful information. Compared with existing studies, our approach differs from them by approaching HMD through modeling task-specific correlation information rather than straightforwardly fusing and matching visual and text features. Particularly, the design of memory and self-rejection training provides an effective learning and optimization solution for such correlation information, showing their potential of being applied to a series of tasks with different nature of describing images such as captioning.

## 6 CONCLUSION

In this paper, we propose an LLM-driven approach for HMD with cross-modal memorizing and self-rejection training, which learns and enhances the task-specific correlation information between visual and text features that result in hateful memes. Experimental results on English and Chinese benchmark datasets confirm the validity of the proposed approach, which outperforms strong baselines and existing studies and achieves state-of-the-art performance. Analyses further show that the combination of memory and self-rejection training demonstrates their superiority in learning such correlation between multimodalities, thereby proves the effectiveness of our model design for HMD.

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

| | HMC | | | | Memeplate | | | |
|---|---|---|---|---|---|---|---|---|
| | Train | Dev | Test | All | Train | Dev | Test | All |
| # of Meme | 8,500 | 500 | 1,000 | 10,000 | 3,746 | 700 | 738 | 5,184 |
| Avg. Tokens Per Meme | 11.7 | 10.2 | 10.4 | 11.5 | 20.3 | 20.4 | 20.0 | 20.3 |

Table 8: Statistics of experiment datasets, where the number of meme and the average number of tokens (i.e., words for English and characters for Chinese) for each meme are reported.

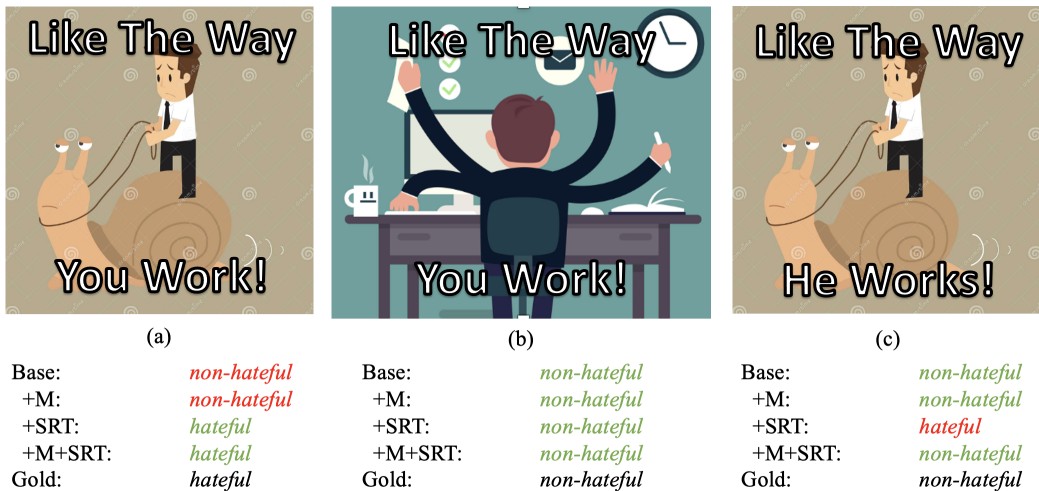

| | (a) | | (b) | | (c) |
|---|---|---|---|---|---|
| Base: | *non-hateful* | Base: | *non-hateful* | Base: | *non-hateful* |
| +M: | *non-hateful* | +M: | *non-hateful* | +M: | *non-hateful* |
| +SRT: | *hateful* | +SRT: | *non-hateful* | +SRT: | *hateful* |
| +M+SRT: | *hateful* | +M+SRT: | *non-hateful* | +M+SRT: | *non-hateful* |
| Gold: | *hateful* | Gold: | *non-hateful* | Gold: | *non-hateful* |

Figure 5: Three memes and predictions of different models on them, with the gold standards also presented. Correct and incorrect labels are highlighted in green and red colors, respectively.

## APPENDIX A: THE STATISTICS OF THE DATASETS

The statistics of the datasets are reported in Table 8, where the number of meme and the average number of tokens (i.e., words for English and characters for Chinese) for each meme are reported.

## APPENDIX B: ADDITIONAL CASE STUDY

In figure 5, we present another group of three memes where our approach is able to make correct predictions, whereas other models fail to do so.

