# OpenReview forum: "LLM-driven Hateful Meme Detection via Cross-modal Memorizing and Self-rejection Training"
_ICLR.cc/2024/Conference — ICLR 2024 Conference Withdrawn Submission_

### Official Review · Reviewer_UptH · 2023-10-23

**Soundness:** 3 good
**Presentation:** 3 good
**Contribution:** 2 fair
**Rating:** 5
**Confidence:** 4

**Summary:**

The paper proposes the use of multimodal Large Language Models in combination with a memory module and a self-rejection training approach for improving the state-of-the-art in the task of Hateful Meme Detection. The authors present in detail their approach and detailed experimental results on two datasets, demonstrating improved performance over a number of SotA methods.

**Strengths:**

- The proposed methodology appears to be sound and well-justified, and it leverages a number of recent advances in multimodal LLMs in order to reach new SotA results on the task of hateful meme detection.
- The presentation is in general very good, including both the writing of the paper and the provided figures.
- The experimental methodology is sound, including comparison to SotA and ablation studies.

**Weaknesses:**

- The very narrow scope of the addressed task raises some doubts about the fit of this work to a venue like ICLR.
- The experiments are limited to only two datasets.
- There are some key points that would require more elaboration (cf. Questions below).

**Questions:**

Nicely described context of the problem and the field's current approaches in abstract and introduction. Also, the paper's contribution is well-motivated but not concretely declared as bullet points in the last part of introduction (probably due to space restrictions) which would be very helpful to several readers.

The methodology is clearly presented and the notation is suitable. I would only demand more rigor through providing the dimensionality of all symbols.
Q1: Why perform random sampling to produce x_m? Wouldn't be the same to average the N' first vectors? Please motivate your choice.
Q2: In rejection sampling, it is unclear why should the average memory vector be equal to the vector of highest reward. Maybe I have missed something.
Answer to Q1 & Q2: The randomness is necessary to obtain x_m^*. The process is done for each (I,X) pair, so after it the best memory vector is obtained.
These questions should be answered before the reader gets confused.

Too few datasets are used for evaluation: only two while there are several meme datasets (Memotion7k, MultiOFF, Harm-C, Harm-P, etc.

Can one gain any explainability with regards to the memory module? What does it actually learn? It seems like a black box that has been named memory module and untenably attributed with correlation-extracting functionality.

The related work is somehow poor. LLMs are missing and the meme literature while sufficient is described only in very high level.

No limitations of this work are provided, including for instance the compute requirements both for training and inference compared with the competing SotA methods for the task.

Minor issue with respect to terminology: I suggest to use multimodal LLMs instead of plain LLMs to avoid confusion.

---

> ### Author Response · Authors · 2023-11-21
> **Response**
>
> Thank you so much for your detailed comments and suggestions; we address the key concerns in the following texts and will include necessary supplements and modifications in the final version of the paper.
>
> >The very narrow scope of the addressed task raises some doubts about the fit of this work to a venue like ICLR.
>
> Regarding the concerns of the scope of the paper, our focus on HMD in a scenario where text and image do not necessarily match presents a unique and interesting challenge. This deviates from conventional image-captioning tasks, bringing a distinct complexity to our work. Our approach is specifically designed to model this complexity, **addressing the particular task-driven relationships between texts and images**. Furthermore, **hateful content is a widespread phenomenon on social media, making the task's scope significant and relevant**. Existing methods of image-text matching or multimodal content understanding often struggle with such tasks, highlighting the importance and novelty of our approach. **We also test our improved model on other datasets and tasks after the paper submission and achieve promising results**, which will be released in later studies.
>
>
> > Too few datasets are used for evaluation: only two while there are several meme datasets (Memotion7k, MultiOFF, Harm-C, Harm-P, etc.
>
> Regarding the concern about the datasets, it is important to emphasize that **these two datasets are widely used benchmarks in the field, especially that the English one serves as the most authentic and only dataset used in many existing studies**. We use them to ensure the reliability and generalizability of our results, where conducting experiments on less popular or unverified datasets might lead to less credible outcomes. Furthermore, **our study specifically targets the most extensively used languages on social media—English and Chinese**. The promising results achieved by our approach on these datasets demonstrate the effectiveness of our approach to processing different languages. However, we will also consider including results from more appropriate datasets and even tasks to further demonstrate the effectiveness of this approach.
>
>
> > The paper's contribution is well-motivated but not concretely declared as bullet points in the last part of introduction (probably due to space restrictions) which would be very helpful to several readers.
> > The methodology is clearly presented and the notation is suitable. I would only demand more rigor through providing the dimensionality of all symbols. Q1: Why perform random sampling to produce x_m? Wouldn't be the same to average the N' first vectors? Please motivate your choice. Q2: In rejection sampling, it is unclear why should the average memory vector be equal to the vector of highest reward. Maybe I have missed something. Answer to Q1 & Q2: The randomness is necessary to obtain x_m^*. The process is done for each (I,X) pair, so after it the best memory vector is obtained. These questions should be answered before the reader gets confused.
> > The related work is somehow poor. LLMs are missing and the meme literature while sufficient is described only in very high level.
> > Minor issue with respect to terminology: I suggest to use multimodal LLMs instead of plain LLMs to avoid confusion.
>
> Thanks for the suggestions on the paper presentation, including the way to present our contributions, the way to obtain x_m^*, more discussions on related work, as well as the terminology of multimodal LLMs, we will address them in the final version of the paper.
>
> > Can one gain any explainability with regards to the memory module? What does it actually learn? It seems like a black box that has been named memory module and untenably attributed with correlation-extracting functionality.
>
> Regarding the questions about the memory module, it is specifically designed to **capture vital information, namely the correlation between visual and textual features that contribute to the identification of hateful memes**. This is achieved through a memory matrix composed of N vectors, where each vector represents a potential aspect that lead to hateful information.
>
> > No limitations of this work are provided, including for instance the compute requirements both for training and inference compared with the competing SotA methods for the task.
>
> Regarding the concerns about limitations of this work and computation requirements, **we find that our model's resource consumption is similar to those models of comparable scale**. Although our training process incorporates an extra self-rejection process, which is not required during inference, the entire process still allows our approach to run efficiently. Importantly, the self-rejection training is able to be conducted independently of the main model training, meaning it does not impose additional resource demands from the primary pipeline. As for other limitations, we will discuss them in the final version of our paper.

---

> ### Comment · Reviewer_UptH · 2023-11-21
> **Response to authors' answer**
>
> I would like to thank the authors for their thoughtful responses. While I can understand some of their points, it appears that they are only willing to perform some minor revisions to the paper text, thus not addressing my main concern, which is the limited (in terms of datasets) experimental study of the proposed method. Therefore, I maintain my score.

---

> > ### Author Response · Authors · 2023-11-22
> > **Response**
> >
> > Thanks for your follow-up comments!
> >
> > Perhaps there is a misunderstanding between us, what we mentioned in our rebuttal is the reason why we use the two datasets in the current version of this paper by following conventional settings in many previous studies, and we analyze the validity of doing so to show the effectiveness of our approach with only two datasets. In the meantime, we do not deny the possibility to add more datasets into our paper to make it stronger. We will run our approach on datasets of relevant areas, including Memotion7k, MultiOFF, Harm-C, and Harm-P for sentiment detection, offensive and hateful speech detection and thus extend the scope of our paper.
> >
> > Still, thanks to make it clear this is the main concern in your opinion, we will address it and add new experiment results into our paper.

---

> > > ### Comment · Reviewer_UptH · 2023-11-22
> > > **Response to authors**
> > >
> > > While I appreciate the positive attitude of the authors, it is unfortunate that now it is too late in the process to consider the promise of including additional experiments as something that can be ensured. Normally, the authors should have already performed the experiments and included them in an updated version in order for the reviewers to re-assess their scores.

---

### Official Review · Reviewer_xmfW · 2023-10-27

**Soundness:** 2 fair
**Presentation:** 2 fair
**Contribution:** 2 fair
**Rating:** 5
**Confidence:** 2

**Summary:**

This work concentrates on leveraging LLMs to address the challenges of indentifying hateful information from the complementarities or contradictions between image and text. The main network follows encoder-decoder paradigm. A memory module with visual and text features is taken as encoder, and the LLM is the decoder for predicting HMD labels. Both of them are trained with the self-rejecting algorithm.

**Strengths:**

Taking LLM as the decoder to generate labels is novel for HMD. The authors have clearly presented their motivation and architectures. The experimental results demonstrate that the proposed methods outperform previous works.

**Weaknesses:**

Although taking LLM for HMD is interesting, the overall methods somewhat lack novelty.

1)The memory module is designed as a matrix. As there is no introduction about its optimization, I assume they are updated in each iteration by BP. Therefore, the modules suffer from limited novelty. In addition, current introduction cannot reflect why they are memories? What does the matrix memorized? The operations in Cross-modal Memorizing looks like attention mechanisms with visual and text features as inputs.

2)The authors should discuss differences between self-rejection training and previous methods. Current introduction makes it looks like standard contrastive learning。

Other issues:

1)From Fig. 1 and Fig. 4, it is insufficient to observe complementarities or contradictions between image and text. The authors should present more examples to demonstrate effectiveness.

2)Fig. 2 is somewhat complicated and confusing. I suggest the authors present HMD and self-rejection training by different figures.

3)The experiments can be further improved. First, the baseline performance without LLM should present in main manuscript. Second, the right column of Table 3 should be removed. The improvements seem minor. Third, from Table 5, it seems that memory module is actually co-attention mechanism. There are only ablation study for the proposed methods. The authors should present some analyse experiments, e.g., compare self-rejection training with similar methods, influences caused by different prompts.

**Questions:**

Please address issues in the Weakness, especially issues of the novelty and experiments.

---

> ### Author Response · Authors · 2023-11-21
> **Response**
>
> Thank you so much for your detailed comments and suggestions; we address the key concerns in the following texts and will include necessary supplements and modifications in the final version of the paper.
>
> >Although taking LLM for HMD is interesting, the overall methods somewhat lack novelty.
> 1)The memory module is designed as a matrix. As there is no introduction about its optimization, I assume they are updated in each iteration by BP. Therefore, the modules suffer from limited novelty. In addition, current introduction cannot reflect why they are memories? What does the matrix memorized? The operations in Cross-modal Memorizing looks like attention mechanisms with visual and text features as inputs.
> 2)The authors should discuss differences between self-rejection training and previous methods. Current introduction makes it looks like standard contrastive learning
>
> Regarding the concerns of novelty, we propose **a reinforcement learning-manner** approach via self-rejection training, which enhances our approach by **learning the task-specific information** and guides it to better understand the relationships between text and images in HMD. To our knowledge, this is the first study that adopt LLM to leverage the capability of LLM and combine it with task-specific information through self-rejection training. Additionally, **we utilize generative large language models (LLMs) for understanding tasks**, a domain where LLMs generally are hard to perform well, especially with "unmatching" text and image pairs. This application showcases the innovativeness of this work, leveraging LLMs in a challenging and novel context.
>
> >From Fig. 1 and Fig. 4, it is insufficient to observe complementarities or contradictions between image and text. The authors should present more examples to demonstrate effectiveness.
>
> Regarding the concern about Figures 1 and 4, in addition to the figures in the current paper, **we prepare a more comprehensive set of examples and plan to include them in the final version of the paper**. These additional examples better illustrate the intricacies of how our approach effectively discerns the complementarities and contradictions between the visual and textual components of memes.
>
> >Fig. 2 is somewhat complicated and confusing. I suggest the authors present HMD and self-rejection training by different figures.
>
> Thanks for the suggestions on Figure 2, we will improve it in the final version of the paper.
>
> >The experiments can be further improved. First, the baseline performance without LLM should present in main manuscript.
>
> Regarding the concerns for the experiments, for the baseline performance without LLM, we do not include it in the main manuscript due to space limitation. We will add it to the  paper in its final version.
>
> >Second, the right column of Table 3 should be removed. The improvements seem minor.
>
> For Table 3, thanks for your suggestions, we will make necessary adjustments. For the improvement, we assess the significance of the improvements via t-test at p<0.05 level and **find them are significant**. Meanwhile, **such performance gain is similar in previous studies**, which adequately proves the effectiveness of our approach.
>
> >Third, from Table 5, it seems that memory module is actually co-attention mechanism. There are only ablation study for the proposed methods. The authors should present some analyse experiments, e.g., compare self-rejection training with similar methods, influences caused by different prompts.
>
> For the differences between the memory module and co-attention mechanism, **the memory module in our approach is designed to specifically capture correlations between visual and textual features in hateful memes**. The memory module includes a list of memory vectors, each targeting a potential aspect of hateful content. In contrast, co-attention mechanisms, commonly used in tasks like image captioning, **focus on aligning and fusing multimodal features to model shared semantics**. This fundamental difference in focus and function results in our memory module being more effective for the specific task of HMD.
>
> For more ablation studies, since our approach emphasizes learning task-specific information, **there are limited existing models for a direct comparison with our self-rejection training approach**. Nonetheless, we try new improvements after the paper submission and achieve promising results, which will be released in later studies.

---

### Official Review · Reviewer_uMip · 2023-10-31

**Soundness:** 3 good
**Presentation:** 3 good
**Contribution:** 3 good
**Rating:** 6
**Confidence:** 4

**Summary:**

In this paper, the authors present a novel approach to detecting hateful memes using LLMs.

**Strengths:**

The paper is well-written and well-motivated. Detecting hateful memes is a challenging task. The authors present a novel LLM-driven approach to detecting hateful memes. They evaluate their proposed approach on two benchmark datasets (in English and Chinese).

**Weaknesses:**

There are a few typos and grammatical mistakes in the paper. E.g., in Table 7 HMC is written as MHC

What might help to further strengthen/motivate such work is to show the effectiveness of off-the-shelf/fine-tuned LLMs in detecting the hatefulness of memes. With no additional components such as cross-model memorization, can we show if LLMs have the capability of predicting the hatefulness of memes?

**Questions:**

See my comment in Weakneses. Additionally, in section 2.2 (reward model training), the method to create the positive and negative samples is unclear. Can we elaborate on it?

Why do we restrict the proposed approach to only detect the hatefulness of memes? Why not use it for a broad range of meme understanding tasks such as harmfulness detection, offensiveness detection, etc.?

---

> ### Author Response · Authors · 2023-11-21
> **Response**
>
> Thank you so much for your detailed comments and suggestions; we address the key concerns in the following texts and will include necessary supplements and modifications in the final version of the paper.
>
>
> >There are a few typos and grammatical mistakes in the paper. E.g., in Table 7 HMC is written as MHC
>
> Regarding typos, thanks for pointing them out. We will address them in the final version of the paper.
>
> >What might help to further strengthen/motivate such work is to show the effectiveness of off-the-shelf/fine-tuned LLMs in detecting the hatefulness of memes. With no additional components such as cross-model memorization, can we show if LLMs have the capability of predicting the hatefulness of memes?
>
> Regarding the concerns about using LLMs in detecting the hatefulness of memes, our base model in the experiments employs a fine-tuned LLM, **which demonstrates the capability of LLMs in hateful meme detection**. However, it is also important to note that while LLMs are capable in this task, **the performance of our proposed approach surpasses that of the base LLM model**. This distinction underlines the enhancements our method offers over the standard LLM approach, thereby reinforcing the significance and contribution of our work.
>
>
> >Additionally, in section 2.2 (reward model training), the method to create the positive and negative samples is unclear. Can we elaborate on it?
>
> Regarding the creation of positive and negative samples for reward model training in Section 2.2 of the paper, the process is described as follows.
> **Positive Samples**: these are **directly selected from the training data**. An instance, which is an image-text pair (denoted as I_r ,X_r ), is randomly chosen from the training data and treated as a positive sample. This means that **the instance is already associated with HMD**, making it relevant for the task.
> **Negative Samples**: to create negative samples, **a caption $C_r$ is generated for the image in the positive sample instance**. This caption is then combined with the image $I_r$ to form a new instance ($I_r$ ,$C_r$), which serves as the negative sample. This approach assumes that the newly generated caption, when paired with the image, **does not carry the hateful meme context**, thus making it irrelevant or negative for the HMD task.
>
>
> >Why do we restrict the proposed approach to only detect the hatefulness of memes? Why not use it for a broad range of meme understanding tasks such as harmfulness detection, offensiveness detection, etc.?
>
> Regarding the application of our proposed approach to HMD, it's primarily driven by the unique challenge presented by this particular scenario. Different from standard image-captioning tasks where there is a direct match between the text and image, **hateful meme analysis often involves dealing with content where the text and image may not necessarily align**. This mismatch brings a distinctive barrier to the task, **making it an interesting and challenging task**. Our approach is specifically designed to model this unique aspect of image-text pair. While this paper emphasizes HMD, it is important to note that **our approach is not limited to this task** alone. We conduct experiments in harmfulness detection in Chinese and report the results in our paper, which demonstrate the effectiveness of our approach in other scenarios as well. Additionally, we test improved versions of our approach on other datasets and tasks and plan to report the results later. Since the content of these studies is still undergoing, we do not to include those results in the discussion.

---

### Official Review · Reviewer_uTqz · 2023-10-31

**Soundness:** 3 good
**Presentation:** 2 fair
**Contribution:** 3 good
**Rating:** 5
**Confidence:** 4

**Summary:**

In this paper, the authors present a memory-based multimodal approach with LLM to address the Hateful Meme Detection(HMD) task. It introduces a self-rejection training strategy to update the memory vectors so it represents the semantic space for the HMD task.

**Strengths:**

The paper introduces a novel method to tackle the HMD task, which can be seamlessly integrated with any advanced LLM to enhance performance.
The paper achieves the SOTA performance on two distinct datasets, each in different language (English and Chinese).

**Weaknesses:**

The writing lacks clarity:

Sec 2.2: Rejection Sampling, ‘we select the correlation vector with the highest reward score’. Where the concept of ‘reward score’ is introduced without an accompanying explanation. It seems being crucial for comprehending of the work.

As I understand, SRJ is introduced to train the memory vectors, there is no clarification regarding the baseline ‘+SRJ’ without memory module.
It would be helpful if the author could also provide an explanation for ‘+M’. What is the ablation way to update memory vectors without SRJ?

What are the value of N’ and M in Sec 2.1 :: Cross-modal Memorizing and the value of ‘T’ in Sec 2.2::Rejection Sampling. Is there an impact on the efficiency of the proposed approach based on these values?

Within the cross-modal memorizing module, the direct concatenation of visual and text embeddings for comparison with the memory vectors may not efficiently capture cross-modal information. Therefore, the size of the memory vectors, M, becomes crucial. It would be beneficial to include an ablation study on the memory vector size to gain further insights.

In the HMC dataset, 10% are unimodal hateful examples. So the <image, caption> pair could be hateful, do you have insights on why this may not adversely affect the model’s performance?


Minor issues:
1). The Figure 2 can be improved by adding explanation for the various arros and lines at the bottom. And consider using different color to distinguish between different modules.
2). Typo in Table 7, MHC→ HMC.
3). For the HMC dataset, the label for each split have been made available at: https://hatefulmemeschallenge.com/# . Additionally,  the ‘Test’ column in Table 3 could be removed if the authors cannot find the GT labels.

**Questions:**

This could be an important piece of work. However, I would appreciate further clarification from the authors regarding the mentioned weakness in above.

---

> ### Author Response · Authors · 2023-11-21
> **Response**
>
> Thank you so much for your detailed comments and suggestions; we address the key concerns in the following texts and will include necessary supplements and modifications in the final version of the paper.
>
> > Sec 2.2: Rejection Sampling, ‘we select the correlation vector with the highest reward score’. Where the concept of ‘reward score’ is introduced without an accompanying explanation. It seems being crucial for comprehending of the work.
>
> Regarding the term "reward score," **it is synonymous with "score" in this context**. This terminology aligns with the usage in LLaMA-2. We will explain this in the final version of the paper.
>
> > As I understand, SRJ is introduced to train the memory vectors, there is no clarification regarding the baseline ‘+SRJ’ without memory module. It would be helpful if the author could also provide an explanation for ‘+M’. What is the ablation way to update memory vectors without SRJ?
>
> Regrading the concerns about the details of the baseline, for "+SRT", we directly use self-rejection sampling to enhance the base models by concatenating visual and text features (i.e., $x_{vt}$) to form the correlation vector (i.e., $x_{m}$), and randomly set 33% values in $x_{m}$ to zero to facilitate self-rejection training. And for "+M", it uses visual information encoding, cross-modal memorizing, and LLM prompting (i.e., the process in Section 2.1) and **the memory is optimized by the cross-entropy loss from comparing the model prediction and the gold standard for the task**.
>
> > What are the value of N’ and M in Sec 2.1 :: Cross-modal Memorizing and the value of ‘T’ in Sec 2.2::Rejection Sampling. Is there an impact on the efficiency of the proposed approach based on these values?
>
> Regarding the impact of variables N', M, and T, we configure N' to 100 for the HMC dataset and 75 for the Memeplate dataset. For both M and T, we consistently set their values to 20 and 4, respectively, across both datasets. Concerning efficiency, as these values are relatively small and the time complexity of performing sampling is O(1), **we do not notice any substantial increase in the running time of our model**.
>
> >Within the cross-modal memorizing module, the direct concatenation of visual and text embeddings for comparison with the memory vectors may not efficiently capture cross-modal information. Therefore, the size of the memory vectors, M, becomes crucial. It would be beneficial to include an ablation study on the memory vector size to gain further insights.
>
> Regarding the ablation on the memory vector size, we conduct experiments to assess how varying the number of memory vectors (denoted as N) affects performance and present the results in Section 4.2. We observe that **with a small N, the model's performance enhances as N increases**. However, **once N surpasses a certain threshold, further increases in N yield negligible improvements**. We run similar experiments with variable M and **obtain similar results**.
>
>
> > In the HMC dataset, 10% are unimodal hateful examples. So the <image, caption> pair could be hateful, do you have insights on why this may not adversely affect the model’s performance?
>
> Regarding the impact of unimodal hateful examples, since **their presence in the dataset is limited** (only 10%), the potential negative effects are not much. Since the majority of the data (90%) is not in unimodal hateful cases, it reduces the overall impact of unimodal hateful content.
>
>
> > Minor issues: 1). The Figure 2 can be improved by adding explanation for the various arros and lines at the bottom. And consider using different color to distinguish between different modules. 2). Typo in Table 7, MHC→ HMC. 3). For the HMC dataset, the label for each split have been made available at: https://hatefulmemeschallenge.com/# . Additionally, the ‘Test’ column in Table 3 could be removed if the authors cannot find the GT labels.
>
> Regarding other issues related to figures, typos, and datasets, thanks for pointing them out. We will address them in the final version of the paper.